# Frequency, Grades of Toxicity, and Predictors of Hepatotoxicity and Acute Kidney Injury with Remdesivir in COVID-19 Patients: A Multicenter Retrospective Cohort Study

**DOI:** 10.3390/healthcare13172143

**Published:** 2025-08-28

**Authors:** Yazed Saleh Alsowaida, Shuroug A. Alowais, Dalal Alsowaida, Alia Alshammari, Bader Alshoumr, Kareemah Alshurtan, Mohammed Almunef, Thamer A. Almangour, Omar A. Alshaya, Khalid Bin Saleh

**Affiliations:** 1Department of Clinical Pharmacy, College of Pharmacy, University of Ha’il, Ha’il 55473, Saudi Arabia; 2Department of Pharmacy Practice, College of Pharmacy, King Saud bin Abdulaziz University for Health Sciences, Riyadh 11461, Saudi Arabia; owaiss@ksau-hs.edu.sa (S.A.A.); omaraalshaya@gmail.com (O.A.A.); binsalehkh@gmail.com (K.B.S.); 3King Abdullah International Medical Research Center, Riyadh 11481, Saudi Arabia; 4Pharmaceutical Care Department, King Abdulaziz Medical City, Riyadh 11481, Saudi Arabia; 5Department of Clinical Laboratory Sciences, College of Applied Medical Sciences, King Saud University, Riyadh 12372, Saudi Arabia; dalsowaida@ksu.edu.sa; 6Department of Pharmaceutics, College of Pharmacy, University of Ha’il, Ha’il 55473, Saudi Arabia; alia.alshammari@uoh.edu.sa; 7Department of Health Informatics, College of Public Health, University of Ha’il, Ha’il 55473, Saudi Arabia; b.alshoumr@uoh.edu.sa; 8Department of Internal Medicine and Adult Critical Care, College of Medicine, University of Ha’il, Ha’il 55473, Saudi Arabia; k.alshurtan@uoh.edu.sa; 9Department of Pharmacy Practice, College of Pharmacy, Qassim University, Buraidah 51452, Saudi Arabia; m.almunef@qu.edu.sa; 10Department of Clinical Pharmacy, College of Pharmacy, King Saud University, Riyadh 12372, Saudi Arabia; talmangour@ksu.edu.sa

**Keywords:** remdesivir, hepatotoxicity, acute kidney injury, liver, COVID-19

## Abstract

**Background**: Remdesivir is associated with hepatotoxicity and acute kidney injury (AKI). The objective of this study was to assess the hepatotoxicity and AKI with remdesivir. **Method:** This is a multicenter, retrospective cohort study for adult patients who used remdesivir for COVID-19 from 3/2020 to 10/2021. The study was conducted at Rhode Island Hospital, Rhode Island, United States. Data were analyzed with descriptive statistics as well as logistic regression analysis using STATA 18. **Results:** A total of 1635 patients were evaluated for hepatotoxicity: 337 developed hepatotoxicity, and 1298 had normal hepatic function. The overall median frequency of hepatotoxicity occurred in 319 patients (19.5%). Patient age (OR 1.02, 95% CI: 1–1.04, *p* = 0.02) and selective serotonin reuptake inhibitors (SSRIs) use (OR 1.7, 95% CI: 1.1–2.6, *p* = 0.01) were potential risk factors for remdesivir-associated hepatotoxicity. In contrast, being male gender was protective against remdesivir-associated hepatotoxicity (OR 0.63, 95% CI: 0.47–0.87, *p* = 0.02). The frequency of AKI with remdesivir occurred in 280 patients (17.3%). **Conclusions:** The frequency of hepatotoxicity was 19.5%, and the frequency of AKI was 17.3%. Increasing age and using SSRIs were risk factors for remdesivir-associated hepatotoxicity, while male gender was a protective factor. Clinicians should vigilantly monitor hepatic and renal functions for patients using remdesivir, especially in elderly patients.

## 1. Introduction

The coronavirus disease 2019 (COVID-19), caused by the severe acute respiratory syndrome coronavirus 2 (SARS-CoV-2), has been a global health challenge since its emergence in 2019 [1]. Although the burden of severe illness has declined due to widespread vaccination, natural immunity, and evolving viral variants, COVID-19 remains an ongoing concern as of 2025. Healthcare systems worldwide continue to manage new cases, particularly among vulnerable populations, and the need for effective antiviral therapies remains critical to improve patient outcomes [2].

Remdesivir, a viral RNA-dependent RNA polymerase inhibitor, demonstrated in vitro activity against SARS-CoV-1 [3]. Due to its antiviral potential, it was quickly recognized as a promising treatment option for COVID-19 due to its ability to inhibit SARS-CoV-2 replication in laboratory studies [3,4]. Based on multiple clinical trials, the U.S. Food and Drug Administration (FDA) approved remdesivir in October 2020 for the treatment of COVID-19 in hospitalized adults and pediatric patients aged 12 years or older who weigh at least 40 kg [5]. The recommended dosage is 200 mg intravenously (IV) on day 1, followed by 100 mg IV daily for 5 to 10 days. It has also been incorporated in the National Institutes of Health (NIH) COVID-19 management guidelines [6].

In clinical settings, concerns remain regarding the safety of remdesivir, particularly in critically ill patients [5,7]. Commonly reported adverse effects include nausea, vomiting, bradycardia, elevated liver enzymes, and AKI, with some studies highlighting a higher prevalence of AKI among older adults with comorbidities. While several investigations showed favorable clinical outcomes, the overall safety and efficacy profile of remdesivir remains mixed and continues to warrant close monitoring and further study [8]. Additionally, a clinical study assessing patients with creatinine clearance (CrCl) <30 mL/min found that the risk of AKI at the end of treatment is present regardless of the baseline CrCl [9,10]. A study by Kim et al. evaluated the hepatobiliary ADR associated with remdesivir; however, it has a limitation in that it did not evaluate the risk factors of hepatotoxicity [11]. Moreover, Li et al. evaluated AKI with remdesivir in COVID-19 patients and found a risk of AKI with remdesivir, but they did not evaluate the grades of AKI [12].

The exact mechanisms by which remdesivir induces liver or kidney injury remain unclear and are difficult to study [13]. However, kidney injury is believed to be related to the accumulation of its solubility enhancer, sulfobutylether β-cyclodextrin sodium (SBECD), in the renal system. In contrast, liver injury is more likely due to direct hepatocellular toxicity caused by remdesivir’s inhibition of mitochondrial RNA polymerase [14]. As a prodrug, remdesivir is metabolized by CYP3A4 and transported by P-glycoprotein, both of which are subject to drug interactions that may increase remdesivir plasma levels. It is worth mentioning that SSRI use was associated with a higher frequency of liver injury, which could be related to their effect on increasing remdesivir plasma levels. Additionally, remdesivir inhibits carboxylesterase-2, an enzyme involved in the metabolism of various drugs and toxins, potentially contributing to increased hepatotoxicity.

Liver injury associated with remdesivir typically presents as elevated AST, ALT, and serum bilirubin or alkaline phosphatase levels, and these abnormalities usually resolve upon drug discontinuation [14,15]. It is also important to consider that renal and hepatic injuries may be partly attributable to the inflammatory response associated with COVID-19 or the effects of other concurrently administered therapies, rather than remdesivir alone [16].

Currently published studies have limitations, including small sample sizes, failure to evaluate the grades of toxicities, or failure to assess potential risk factors. Therefore, the objective of this study is to assess the frequency, grades of toxicity, and risk factors for hepatotoxicity and AKI associated with remdesivir in patients hospitalized with COVID-19.

## 2. Materials and Methods

### 2.1. Study Design and Setting

This is a multicenter, retrospective cohort study for patients who used remdesivir (Gilead, Foster City, CA, USA) for COVID-19 from March 2020 to October 2021. The study was conducted at Rhode Island Hospital and the Miriam Hospital in Providence, RI, USA.

### 2.2. Included and Excluded Patients

Adult patients (≥18 years old) who were diagnosed with COVID-19 by the SARS-CoV-2 test and received remdesivir. The remdesivir was administered by a loading dose of 200 mg intravenously on the first day, followed by 100 mg daily by an intravenous route for up to 10 days of treatment based on the physician’s judgment. The timing of toxicity will be considered from the first day of remdesivir administration until 24 h after the last dose of remdesivir to account for remdesivir and its active metabolites [17]. Patients with normal and abnormal baseline hepatic and renal impairment will be included based on the definition below in Section 2.7. Patients with missing hepatic and renal laboratory data will be excluded.

### 2.3. Outcomes

The primary outcome was the frequency of hepatotoxicity and nephrotoxicity. Secondary outcomes were grades of hepatotoxicity and nephrotoxicity, and risk factors for hepatotoxicity. Notably, outcome thresholds for hepatotoxicity were based on Common Terminology Criteria for Adverse Events (CTCAE), version 5 [18]; outcomes for nephrotoxicity were based on the CACTE and the KDIGO guideline, 2012 [18,19].

### 2.4. Data Collection

The data for each patient was extracted from the Electronic Health Record and exported as an Excel sheet for statistical analysis. The following data were obtained: patients’ demographic information, laboratory results, vital readings, disease comorbidities, and treatment settings.

### 2.5. Statistical Analysis

Patients’ baseline results were obtained as mean with standard deviation or median with interquartile range and analyzed with the Student’s *t*-test and the Mann-Whitney-Wilcoxon test, respectively. Categorical variables were analyzed with Pearson’s chi-square test.

We estimated the frequency of hepatotoxicity based on the median percentage for hepatic biomarker enzyme elevation of aspartate aminotransferase (AST), alanine aminotransferase (ALT), and bilirubin.

Risk factors were analyzed using univariate and multivariate logistic regression with the adjustment of known confounders. We selected covariates for multivariate logistic regression based on diseases, laboratory abnormalities, and hepatotoxic and nephrotoxic drugs based on disease pathophysiology and published studies. Specifically, age, gender, race, drugs associated with hepatotoxicity, diabetes, hypertension, and baseline liver function tests. A statistically significant test was determined at a *p*-value threshold of 0.05. Statistical analysis was performed using Stata, version 18 (Stata Corporation, College Station, TX, USA).

### 2.6. Power Calculation for Sample Size

The sample was estimated based on a two-tailed test with an effect size of 10%, a probability of error (*alpha*) of 5%, and a power of 95%. The test yielded a minimum sample size of 1289 patients with a critical t value of 1.96. Notably, the effect size is estimated based on the literature with estimates approximately ranging from 10% [11]. The power calculation was performed using G*Power software version 3.1 (©2025 Heinrich-Heine-Universität Düsseldorf).

### 2.7. Definitions

Hepatotoxicity: elevation of AST or ALT more than 3 times the upper limit of normal if the baseline was normal, or 1.5–3.0× if the baseline was abnormal; elevation of bilirubin more than 1.5× the upper limit of normal if the baseline was normal, or more than 1.0–1.5× if the baseline was abnormal [18].Acute kidney injury: Increase in serum creatinine (Scr) by ≥0.3 mg/dL (≥26.5 µmol/L) within 48 h; or increase in SCr to ≥1.5 times baseline occurring within 7 days; or Urine volume of <0.5 mL/kg/h for 6 h [19].Grade 1 toxicity: Asymptomatic or mild symptoms without the need for intervention [18].Grade 2 toxicity: Moderate symptoms that need local or noninvasive intervention [18].Grade 3 toxicity: Severe or significant but not life-threatening, which warrants hospitalization or prolongation of the hospitalization course [18].Grade 4 toxicity: Results in life-threatening implications that warrant urgent intervention [18].Grade 5 toxicity: death related to the adverse drug reaction (ADR) [18].

## 3. Results

Overall, 1635 patients were evaluated for hepatotoxicity: 337 developed hepatotoxicity, and 1298 had normal hepatic function. Patients with normal hepatic function had a significantly higher mean age of 68 versus 63.2 for patients with hepatotoxicity, *p* < 0.001, with significantly more male patients with hepatotoxicity (65%), *p* < 0.001. There were no significant differences in race distribution for patients who developed hepatotoxicity vs. those with normal hepatic function, *p* = 0.3. The most common comorbidities were diabetes (31.1%), hypertension (60%), obesity (27.5%), and cardiac arrhythmias (18.3%). The presence of hepatic diseases was not statistically different between groups, *p* = 0.38. Complete baseline characteristics are available in Table 1.

### 3.1. The Frequency and Grades of Remdesivir-Associated Hepatotoxicity

The overall median frequency of hepatotoxicity occurred in 319 patients (19.5%). To elaborate, the frequency of AST elevation occurred in 319 patients (19.5%), with the majority of patients developing grade 1 AST elevation (262; 16.5%), the frequency of ALT elevation occurred in 337 patients (20.6%), with most patients developing grade 1 ALT elevation (281; 17.7%), and the frequency of bilirubin elevation occurred in 122 patients (7.5%), with the majority of patients with grade 1 bilirubin elevation (40; 2.5%). Grades of hepatotoxicity are available in Figure 1.

We did not find differences in dexamethasone use among patients with hepatotoxicity (99.4%) versus those with normal liver function (98.3%), *p* = 0.14. Lastly, there were significantly fewer patients with hepatotoxicity with no ventilator, 44 (13.8%), versus 70 (5.3%) in patients with normal liver function, *p* < 0.001. Patients with hepatotoxicity have a statistically significantly longer mean duration of hospitalization, 10.3 days (SD 10.5) versus 7.6 (SD 7.7) for patients with normal liver function, *p* < 0.001. There were statistically significantly fewer patients requiring non-invasive positive pressure ventilation in the hepatotoxicity group, 138 (43.3%) versus 443 (33.7%) in patients with normal liver function, *p* = 0.001.

### 3.2. Predictors of Remdesivir-Associated Hepatotoxicity

After adjusting the logistic regression model, hepatotoxicity with remdesivir use was associated with patient age (OR 1.02, 95% CI: 1–1.04, *p* = 0.02) and selective serotonin reuptake inhibitors (SSRI) use (OR 1.7, 95% CI: 1.1–2.6, *p* = 0.01). Being male and ventilator use were negatively associated with hepatotoxicity with remdesivir (OR 0.63, 95% CI: 0.46–0.87, *p* = 0.005) and (OR 0.24, 95% CI: 0.06–0.86, *p* = 0.02), respectively. Risk factors of remdesivir-associated hepatotoxicity are available in Table 2.

### 3.3. The Frequency and Grades of Remdesivir-Associated Acute Kidney Injury

The frequency of AKI with remdesivir use occurred in 280 patients (17.3%). Specifically, 141 patients (8.8%) developed grade 1 AKI, 29 patients (1.8%) developed grade 2 AKI, and 87 patients (5.4%) developed grade 3 AKI. Grades of AKI are available in Figure 1. Patients who developed AKI were on longer durations of remdesivir therapy; 4.5 days vs. 4.27 days in patients who did not develop AKI, OR 0.87 (95% CI: 0.79–0.93, *p* = 0.006). Although the difference is statistically significant, it is clinically insignificant.

## 4. Discussion

Remdesivir is an essential antiviral agent with a broad spectrum of activity against several viruses, including SARS-CoV-2 [3]. Remdesivir was a key player in controlling the COVID-19 pandemic. However, remdesivir does not come without significant toxicities. In this study, we evaluated the frequency and grades of hepatotoxicity and AKI in addition to the predictors of hepatotoxicity. We found the frequency of hepatotoxicity to be 19.5%, with most patients developing grade 1 hepatotoxicity based on ALT, AST, and bilirubin. Increasing age and SSRI use were predictors of hepatotoxicity, while male gender was a protective factor against remdesivir-associated hepatotoxicity. Moreover, the frequency of AKI was 17.3%, with most patients developing grade 1 AKI.

The median frequency of remdesivir-associated hepatotoxicity in our study (19.5%) has similarities and differences to the published literature. A study by Kim et al. evaluated hepatobiliary ADR with remdesivir therapy in COVID-19 patients, which included 2107 patients from VigiBase [11]. The authors found the median frequency of hepatotoxicity was 11.2%, which is lower than the estimates in our study, which could be explained by several factors. First, the study included ADRs since the inception of VigiBase; therefore, the study may have included indications other than remdesivir. In contrast, in our study, we included only COVID-19 patients. Thus, our study could provide more accurate estimates of remdesivir-associated hepatotoxicity in COVID-19 patients. In a randomized controlled trial, ACCT-1 evaluated the efficacy and safety of remdesivir in hospitalized COVID-19 patients and found the liver function test elevations to be 6% in both the remdesivir and placebo groups [3]. The difference compared to our study is that they did not specify the types of enzymes elevated, and there is a risk of bias. Lastly, a study by Wong et al. evaluated the risks of hepatotoxicity with remdesivir in COVID-19 patients and included 860 patients [21]. The study found the frequency of hepatotoxicity was 38.8%. The estimate of hepatotoxicity was higher compared to our study of 19.5%, perhaps due to the smaller sample size in the study by Wong et al., since it included 860 patients. Another study by Jinda et al. that evaluated hepatotoxicity from remdesivir in pediatric patients found that the frequency of hepatotoxicity was 18.2% [22]. Notably, the study has a limited sample size of only 66 patients. Lastly, a study by Seyedalipour et al. conducted in Iran to evaluate the hepatotoxicity of remdesivir found ALT and AST increased by 6.20 and 3.64 times, respectively [23]. From our study and other studies’ findings, we believe the frequency of remdesivir hepatotoxicity is 20–30%.

Discussing specific liver enzymes, the abnormalities in liver enzymes in our study are comparable to other studies. Specifically, ALT was elevated in 20.6% of the patients, which is relatively close to estimates by Kim et al. of 17.6% [11]. In contrast, our finding of AST elevation was higher than that reported in the study by Kim et al.; our study found the frequency of AST elevation to be 19.5%, while the frequency of AST elevation in the study by Kim et al. was 11.2%. Ultimately, the bilirubin level elevation in our study was 7.5%, which contradicts the findings of the study by Kim et al., since they found 1.5% bilirubin elevation. Differences compared to our findings can be explained by several factors, including sample size and patient population, male gender, obesity, and truncal obesity [24]. Notably, some factors are not associated with COVID-19; still, it can impact the bilirubin level.

Among patients who developed hepatotoxicity, the majority of hepatic enzyme biomarkers (ALT, AST, and bilirubin) were grade 1 (mild hepatic impairment). Our findings have similarities to the study by Seyedalipour et al., since they found that the majority of the hepatotoxicity was grade 1 and grade 2, with OR for ALT 5.30 (95% CI: 3.39–8.27, *p* < 0.001), and the OR for AST 3.11 (95% CI: 1.90–5.08, *p* < 0.001) [23]. Similarly, a study by Wong et al. found that 79.6% of patients who developed hepatotoxicity were grade 1 and grade 2 without specifying how many patients were in grade 1 and grade 2 separately [21]. Similar to our study, other studies evaluating specific patients with grades of hepatotoxicity are lacking.

In our study, we evaluated the risk factors of hepatotoxicity with remdesivir use, and we identified 2 risk factors. After adjusting for risk of liver disease, we found that the higher the patient age in years, the higher the risk of remdesivir-associated hepatotoxicity, OR 1.02 (95% CI: 1–1.04, *p* = 0.02). A likely explanation of age as a risk factor is that elderly patients’ livers undergo cellular changes with aging, which make them susceptible to drug-induced hepatotoxicity [25]. Unfortunately, currently, no published studies evaluating the risk factors of remdesivir and hepatotoxicity in adult patients exist to compare against our study. However, a study by Jinda et al. evaluated risk factors of hepatotoxicity with remdesivir use in pediatric COVID-19 patients and found that the longer median duration of remdesivir use (OR 3,95% CI: 0–5, *p* = 0.01) and the higher the dose of remdesivir (OR 5.3, 95% CI: 0.5–10.3, *p* = 0.01) were risk factors for remdesivir-associated hepatotoxicity [22]. Another risk factor for remdesivir-associated hepatotoxicity is the use of SSRIs, since we found the OR of 1.7 (95% CI: 1.1–2.6, *p* = 0.01). Our finding is consistent with a study by Huang et al. that evaluated the risk of liver injury and found that SSRIs increase the odds of liver injury (OR 1.22, 95% CI: 1.16–1.29) [26]. Several types of lesions occur due to SSRI-induced liver injury by immunologic or idiosyncratic reactions, including acute hepatitis, cholestatic hepatitis, steatohepatitis, and transaminitis [27]. Lastly, in our study, male gender was found to be a protective factor against remdesivir-associated hepatotoxicity (OR 0.63, 95% CI: 0.47–0.87, *p* = 0.02). A possible explanation for our findings was that females have a higher risk for drug-induced hepatotoxicity generally [28].

In our study, we evaluated the risk of AKI and found that 17.3% of the patients developed AKI, which is comparable to published studies. A study by Wong et al. found the frequency rate of AKI to be 15.9% [21]. Notably, the estimate of the AKI is robust in our study compared to the study by Wong et al., since we have a bigger sample size of more than 1600 patients. Another study by Li et al. evaluated the risk of AKI with remdesivir in COVID-19 patients using disproportionality analysis and found that remdesivir increases the odds of AKI with an OR of 2 (95% CI: 1.83–2.18) [12]. Regarding the grades of AKI, the majority of the patients developed grade 1 AKI. Currently, no studies have evaluated the grades of AKI to compare against our estimate.

To our knowledge, our study is multicenter and the first to study to identify risk factors of hepatotoxicity as well as estimate the grades of hepatotoxicity and AKI. Despite that, our study has limitations. First, it is a retrospective cohort with the possibility of missing information and the introduction of bias. The study is a retrospective cohort study and may not establish causality, but it can only establish associations related to the concerned research question. Our study was conducted in a single geographical location and may not be generalizable globally due to differences in healthcare systems, genetic profiles, treatment protocols, and concomitant medications. Our study lacks a control group to ascertain the association of hepatotoxicity and AKI with remdesivir; however, we adjusted for drugs that are associated with hepatotoxicity and AKI in our analysis (For example, Mycobacterium tuberculosis drugs, vancomycin, etc.). Some patients who developed hepatotoxicity or AKI do not have information regarding the grade of toxicity due to missing laboratory information. We could not collect information on a specific SSRI type and dose, but we only evaluated the SSRI as a drug class. We could not obtain data regarding the AKI etiologies (pre-renal, intrinsic, or post-renal). There may be unknown confounders that were not measured. We could not retrieve the dosage regimens’ data of concurrent medications. We could evaluate if there was clinical deterioration due to hepatic enzyme elevations or AKI. There were confounders due to complications of COVID-19; however, we performed multivariate logistic regression and adjusted for the potential confounders.

## 5. Conclusions

Remdesivir is an essential antiviral drug for the treatment of several viral illnesses, including COVID-19. Remdesivir is associated with several ADRs, more importantly, hepatotoxicity and AKI. The frequency of remdesivir-associated hepatotoxicity was 19.5%, with the majority of patients developing grade 1 hepatotoxicity. Predictors of hepatotoxicity include increasing age and the use of SSRIs. On the contrary, the male gender was a protective factor against remdesivir-associated hepatotoxicity. To add, the frequency of AKI was 17.3%, with the majority of patients developing grade 1 AKI. Therefore, clinicians should be vigilantly monitoring patients on remdesivir for hepatotoxicity and AKI through periodic assessment of liver function tests and serum creatinine concentrations, especially in elderly patients. Lastly, clinicians should be cautious when combining remdesivir with SSRIs.

## Figures and Tables

**Figure 1 healthcare-13-02143-f001:**
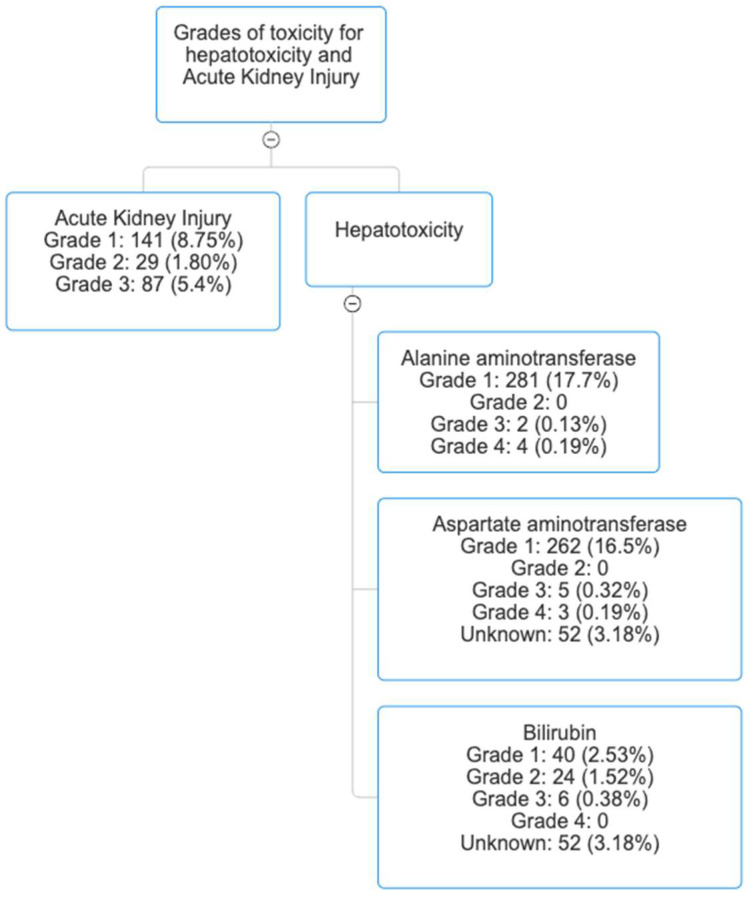
Grades of toxicities for hepatotoxicity and Acute Kidney Injury.

**Table 1 healthcare-13-02143-t001:** Baseline characteristics.

	Total (N = 1635; 100.0%)	Hepatotoxicity (N = 337; 20.6%)	Normal Liver Function (N = 1298; 79.4%)	*p*-Value
Patient age, years, mean (SD)	67.098 (16.184)	63.270 (15.513)	68.092 (16.213)	<0.001
Patient age groups				
18–39 years old 40–64 years old 65 and over	94 (5.7%) 579 (35.4%) 962 (58.8%)	26 (7.7%) 145 (43.0%) 166 (49.3%)	68 (5.2%) 434 (33.4%) 796 (61.3%)	<0.001
Patient Sex				
Female Male	790 (48.3%) 845 (51.7%)	118 (35.0%) 219 (65.0%)	672 (51.8%) 626 (48.2%)	<0.001
Patient race				
Hispanic or Latino Non-Hispanic Black Non-Hispanic White Other/Unknown	295 (18.0%) 155 (9.5%) 1069 (65.4%) 116 (7.1%)	64 (19.0%) 33 (9.8%) 209 (62.0%) 31 (9.2%)	231 (17.8%) 122 (9.4%) 860 (66.3%) 85 (6.5%)	0.303
Non-invasive Positive Pressure Ventilation	581 (35.5%)	134 (39.8%)	447 (34.4%)	0.069
ICU	299 (18.3%)	78 (23.1%)	221 (17.0%)	0.010
Ventilator	114 (7.0%)	46 (13.6%)	68 (5.2%)	<0.001
Comorbidities				
Coronary heart disease	414 (25.3%)	58 (17.2%)	356 (27.4%)	<0.001
Congestive heart failure	189 (11.6%)	21 (6.2%)	168 (12.9%)	<0.001
Cardiac arrhythmias	300 (18.3%)	41 (12.2%)	259 (20.0%)	<0.001
Valvular disease	104 (6.4%)	19 (5.6%)	85 (6.5%)	0.542
Pulmonary circulation disorders	83 (5.1%)	11 (3.3%)	72 (5.5%)	0.089
Peripheral vascular disorders	126 (7.7%)	21 (6.2%)	105 (8.1%)	0.255
Chronic pulmonary disease	379 (23.2%)	56 (16.6%)	323 (24.9%)	0.001
Hypertension	982 (60.1%)	184 (54.6%)	798 (61.5%)	0.022
Diabetes	509 (31.1%)	85 (25.2%)	424 (32.7%)	0.009
Hypothyroidism	147 (9.0%)	25 (7.4%)	122 (9.4%)	0.257
Obesity	450 (27.5%)	79 (23.4%)	371 (28.6%)	0.060
Coagulopathy	78 (4.8%)	14 (4.2%)	64 (4.9%)	0.551
Deficiency anemia	61 (3.7%)	8 (2.4%)	53 (4.1%)	0.140
Renal failure	171 (10.5%)	23 (6.8%)	148 (11.4%)	0.014
Liver disease	64 (3.9%)	16 (4.7%)	48 (3.7%)	0.376
Peptic ulcer disease	22 (1.3%)	1 (0.3%)	21 (1.6%)	0.061
Rheumatoid arthritis	71 (4.3%)	11 (3.3%)	60 (4.6%)	0.276
HIV	8 (0.5%)	3 (0.9%)	5 (0.4%)	0.237
Lymphoma	26 (1.6%)	4 (1.2%)	22 (1.7%)	0.507
Metastatic cancer	55 (3.4%)	8 (2.4%)	47 (3.6%)	0.258
Solid tumor	171 (10.5%)	26 (7.7%)	145 (11.2%)	0.065
Paralysis *	6 (0.4%)	3 (0.9%)	3 (0.2%)	0.075
Neurological disorders	135 (8.3%)	20 (5.9%)	115 (8.9%)	0.082
Depression	297 (18.2%)	46 (13.6%)	251 (19.3%)	0.016
Alcohol abuse	35 (2.1%)	8 (2.4%)	27 (2.1%)	0.740
Drug abuse	27 (1.7%)	4 (1.2%)	23 (1.8%)	0.453
Laboratory parameters				
ALT in unit/Liter, mean (SD) ^$^	6.039 (4.255)	7.189 (5.657)	5.743 (3.759)	<0.001
AST in unit/Liter, mean (SD) ^#^	6.064 (4.264)	7.189 (5.652)	5.775 (3.775)	<0.001
Bilirubin in micromole/Liter, mean (SD) ^^^	5.991 (4.177)	7.120 (5.578)	5.700 (3.679)	<0.001
Creatinine in micromole/Liter, mean (SD) ^&^	9.623 (11.096)	12.231 (13.957)	8.951 (10.130)	<0.001
LOS, mean (SD)	8.390 (8.476)	10.395 (10.515)	7.869 (7.782)	<0.001
Fluid and electrolyte abnormalities	209 (12.8%)	27 (8.0%)	182 (14.0%)	0.003
COVID-19 therapeutics				
Remdesivir duration, days, mean (SD)	4.314 (1.405)	4.318 (1.493)	4.314 (1.382)	0.963
Dexamethasone use	1610 (98.5%)	333 (98.8%)	1277 (98.4%)	0.566
Tocilizumab	27 (1.7%)	11 (3.3%)	16 (1.2%)	0.009
Other concurrent medications				
Mycobacterium tuberculosis drugs	1 (0.1%)	0 (0.0%)	1 (0.1%)	0.612
Antiarrhythmic drugs	47 (3.4%)	18 (6.4%)	29 (2.7%)	0.002
Antifungal agents	38 (2.8%)	10 (3.6%)	28 (2.6%)	0.364
Statins dyslipidemia drugs	818 (59.8%)	154 (55.0%)	664 (61.0%)	0.069
Macrolide antibiotics	224 (16.4%)	60 (21.4%)	164 (15.1%)	0.010
Quinolone antibiotics	77 (5.6%)	20 (7.1%)	57 (5.2%)	0.216
Selective serotonin reuptake inhibitors	300 (21.9%)	42 (15.0%)	258 (23.7%)	0.002
Sympathetic agents	39 (2.8%)	18 (6.4%)	21 (1.9%)	<0.001
Tricyclic antidepressants	42 (3.1%)	6 (2.1%)	36 (3.3%)	0.314
Vancomycin	280 (20.5%)	74 (26.4%)	206 (18.9%)	0.005

Abbreviation: AST: aspartate aminotransferase, ALT: alanine aminotransferase, HIV: human immunodeficiency virus, ICU: intensive care unit. *: hemiplegia and paraplegia, ($: normal ALT level 4–36 u/L, #: normal AST level 8–33 u/L, ^: normal bilirubin level 1.7–20.5 micromole/L, &: normal serum creatinine level 44–97 micromole/L) [20].

**Table 2 healthcare-13-02143-t002:** Risk factors of remdesivir-associated hepatotoxicity.

Variable	Odds Ratio	95% Confidence Interval	*p*-Value
Patient age	1.02	1–1.04	0.02
Male gender	0.63	0.47–0.87	0.005
Race			
Hispanic Black Other	1.07 0.86 0.77	0.7–1.62 0.5–1.45 0.42–1.38	0.7 0.57 0.38
ICU admission	1.9	0.9–4.04	0.09
Length of hospital stay	0.97	0.94–1.01	0.17
Ventilator use	0.24	0.06–0.86	0.02
SSRIs use	1.73	1.13–2.65	0.01
Liver disease *	0.68	0.33–1.38	0.28
ALT	0.84	0.43–1.64	0.61
AST	1.61	0.77–3.35	0.2
Bilirubin	0.7	0.43–1.12	0.13
Serum creatinine	1.01	0.98–1.05	0.2
Duration of remdesivir and hepatotoxicity	0.99	0.91–1.08	0.96

Abbreviation: AST: aspartate aminotransferase, ALT: alanine aminotransferase, AKI: Acute Kidney Injury, ICU: intensive care unit. SSRI: Selective serotonin reuptake inhibitor. *: liver disease based on clinical and laboratory values.

## Data Availability

The data will be made available upon request by the corresponding author.

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
