# Peer review of "Frequency, Grades of Toxicity, and Predictors of Hepatotoxicity and Acute Kidney Injury with Remdesivir in COVID-19 Patients: A Multicenter Retrospective Cohort Study"

_healthcare, 2025, doi:10.3390/healthcare13172143_

Round 1
Reviewer 1 Report
Comments and Suggestions for Authors
Incidence, grades of toxicity, and predictors of hepatotoxicity and acute kidney injury with remdesivir in COVID-19 patients: A multicenter retrospective cohort study
The study was well designed and executed. It was done with an objective to evaluate the incidence, severity (grades), and predictors of hepatotoxicity and acute kidney injury (AKI) associated with remdesivir in hospitalized COVID-19 patients. The article mentions it as a multicenter retrospective cohort study in the setting of Rhode Island Hospital and Miriam Hospital, USA. However, no author is from the said institute. Sample size was 1635 adult patients treated with remdesivir which is a good number for statistical power compared to previous smaller cohorts.
Data has been collected from HER and toxicity grading followed CTCAE and KDIGO guidelines. Logistic regression adjusted for confounders like age, gender, comorbidities, and concurrent drug use, improving the reliability of identified predictors. However, retrospective design may not be able to establish causality – can establish only associations 9low internal validity).
The study was conducted in single geographical region, the findings may not be generalizable globally (differences in healthcare systems, genetic differences and different treatment protocols and concomitant medications). The authors have not provided data of COVID 19 patients who were not treated with remdesivir (lack of control group) which once again limits the ability of attribute hepatotoxicity and AKI solely to remdesivir. Some grades have been marked “unknown” – do they affect accuracy of grading?
The association of hepatotoxicity with SSRI usage - causality is speculative. There is no information on the SSRI type or dosage. The study surely adds value by grading toxicity and identifying predictors, which many previous studies lacked.
The article provides a valid clinical implication: Regular liver and kidney function tests are essential during remdesivir therapy, vigilance when co-prescribing SSRIs with remdesivir and necessity to monitor elderly patients while on remdesivir therapy.

Author Response
Response to Reviewer 1
Dear Editor,
We are pleased to submit our revised manuscript entitled: “Incidence, grades of toxicity, and predictors of hepatotoxicity and acute kidney injury with remdesivir in COVID-19 patients: A multicenter retrospective cohort study.” We would like to thank the reviewers for their positive response to our manuscript. We appreciate the comments and have incorporated their feedback into our revised manuscript. Please find below a point-by-point response to the reviewer’s comments.
Reviewer 1
Comment 1:
The study was well designed and executed. It was done with an objective to evaluate the incidence, severity (grades), and predictors of hepatotoxicity and acute kidney injury (AKI) associated with remdesivir in hospitalized COVID-19 patients. The article mentions it as a multicenter retrospective cohort study in the setting of Rhode Island Hospital and Miriam Hospital, USA. However, no author is from the said institute. The sample size was 1635 adult patients treated with remdesivir, which is a good number for statistical power compared to previous smaller cohorts.
Response 1:
Thank you for your comment. Concerning your comment on authors from Rhode Island Hospital, the first author of the paper/the corresponding author (Yazed Saleh Alsowaida) was a research fellow at Rhode Island Hospital/Merian Hospital from 2021-2022 based on a scholarship agreement contract between Rhode Island Hospital and Saudi Arabia. The researcher conducted a large project on COVID-19 with Rhode Island Hospital approval number # 1589781. The first project was sponsored and funded by Rhode Island Hospital and published with the authors’ affiliation revealing Rhode Island Hospital. The researcher returned to Saudi Arabia as faculty and obtained a research grant from a current employee (University of Hail) that completely funded the secondary analysis project (current project) (Incidence, grades of toxicity, and predictors of hepatotoxicity and acute kidney injury with remdesivir in COVID-19 patients: A multicenter retrospective cohort study). The execution of the second project was a secondary analysis done in Saudi Arabia with the University of Hail. There is a requirement to keep the affiliation of the funding institution, which is why the current affiliation of the first author did not include Rhode Island Hospital.
Comment 2:
Data has been collected from HER and toxicity grading followed CTCAE and KDIGO guidelines. Logistic regression adjusted for confounders like age, gender, comorbidities, and concurrent drug use, improving the reliability of identified predictors. However, retrospective design may not be able to establish causality – can establish only associations 9low internal validity).
Response 2:
Thank you for your comment. We agree that the retrospective cohort study cannot establish causality and only establish associations. That is a limitation of the retrospective cohort studies. We added a sentence in the limitation section of the last paragraph of the discussion section as follows “The study is a retrospective cohort study and may not establish causality, but can only establish associations of the concerned research question.”
Comment 3:
The study was conducted in single geographical region, the findings may not be generalizable globally (differences in healthcare systems, genetic differences and different treatment protocols and concomitant medications). The authors have not provided data of COVID 19 patients who were not treated with remdesivir (lack of control group) which once again limits the ability of attribute hepatotoxicity and AKI solely to remdesivir. Some grades have been marked “unknown” – do they affect accuracy of grading?
Response 3:
Thank you for your comment. We agree with you that our study may not be generalizable globally due to the single geographical location of the study. We will take that into account and acknowledge that in the limitation section as follows “Our study was conducted in a single geographical location and may not be generalizable globally due to differences in healthcare systems, genetic profiles, treatment protocols, and concomitant medications.”
With regard to lacking a control group, we agree that you our study lacks a control group; however, in solutions for that, in our study, we adjusted for drugs that may be associated with hepatotoxicity or AKI, as you can see in Table 1 under the section “Other concurrent medications.” We added that as a limitation as follows: “Our study lacks a control group to ascertain the association of hepatotoxicity and AKI with remdesivir; however, we adjusted for drugs that are associated with hepatotoxicity and AKI in our analysis (For example, Mycobacterium tuberculosis drugs, vancomycin, etc).”
Lastly, some patients who developed hepatotoxicity or AKI have missing grade of toxicity information, and due to the lack of some laboratory tests and unfortunately, that is not retrievable. We added that as a limitation as follows:” Some patients who developed hepatotoxicity or AKI do not have information regarding the grade of toxicity due to missing laboratory information.”
Comment 4:
The association of hepatotoxicity with SSRI usage - causality is speculative. There is no information on the SSRI type or dosage. The study surely adds value by grading toxicity and identifying predictors, which many previous studies lacked
Response 4:
Thank you for your comment. Since it is a retrospective cohort study, we cannot ascertain causality, and it is only an association. We indicated that as a limitation as follows: “The study is a retrospective cohort study and may not establish causality, but can only establish associations of the concerned research question.” Concerning the SSRI type or dosage, the system of data extraction cannot identify specific drugs and/or dosages and was looking at signals based on drug classes since there were > 1600 patients included. We added that as a limitation as follows: “We could not collect information on a specific SSRI type and dose, but we only evaluated the SSRI as a drug class.”
Comment 5:
The article provides a valid clinical implication: Regular liver and kidney function tests are essential during remdesivir therapy, vigilance when co-prescribing SSRIs with remdesivir and necessity to monitor elderly patients while on remdesivir therapy.
Response 5:
Thank you for your comment
Comment 6:
“The incidence” is the term appropriate as this statistically defined term? Many other locations of the manuscript, the reviewer commented on the incidence
Repones 6:
Thank you for your comment. Based on your suggestions, we replaced the word incidence with “frequency” throughout the manuscript.
Comment 7:
Typos and format of p-value
Response 7:
Thank you for your comment. We corrected all the typos with p-value and changed the format of P-value to a smaller p-value, corrected typos, and put percentage values between parentheses () if missing.
Comment 8: “Selective Serotonin Reuptake Inhibitor”, which drug specifically?
Response 8:
Concerning the SSRI type or dosage, the system of data extraction cannot identify specific drugs and/or dosages and was looking at signals based on drug classes since there were > 1600 patients included. We added that as a limitation as follows: “We could not collect information on a specific SSRI type and dose, but we only evaluated the SSRI as a drug class.”
Comment 9: word choice
Response 9:
- We changed the term from “advanced age” to “increasing age” based on your suggestions
- Added “On COVID-19 patients” to the sentence “Thus, our study could provide more accurate estimates of remdesivir-associated hepatotoxicity in COVID-19 patients”
- Changed the term “hepatic enzymes” in conclusion section to “liver function tests”
- We revised the title of Table 2 to “Table 2: Grades of toxicities for hepatotoxicity and Acute Kidney Injury”
- In Table 1: “Heart disease,” we corrected the term and named the variable “Coronary heart disease.”
- In Table 1: “paralysis,” it includes hemiplegia and paraplegia, we added a notation under the Table *: hemiplegia and paraplegia.
- In Table 1: “fluid and electrolyte disorders” we changed the word disorders to “abnormalities”
Comment 10: “Several 229 factors were associated with hepatic enzyme increase, including male gender, obesity, and 230 truncal obesity” This is not in COVID-19 patients as the paper was published in 2018
Response 10: Thank you for your comment. You are correct, this paper is not in COVID-19 patients; however, some factors in this study are associated with increased bilirubin, which this why we cited and included this study in the discussion section. We revised the sentence to avoid confusion as follows:” Differences compared to our findings can be explained by several factors, including sample size and patient population, male gender, obesity, and truncal obesity [22]. Notably, some factors are not associated with COVID-19; still, it can impact the bilirubin level.”
Comment 11: “grade 1” Table 2 mentions values in most grades?
Response 11: Thank you for your comment, we corrected the mistake in this sentence as follows :” The majority of hepatic enzyme biomarkers (ALT, AST, and bilirubin) were grade 1”
Comment 12: in Table 1 “renal failure” and “liver failure”
Response 12: This value represents participants who had renal failure, end-stage renal disease, or end-stage liver disease. Please note that not every patient who develops hepatotoxicity or AKI will have terminal hepatic or liver disease
Comment 13: In Table 1 “Weight loss”
Response 13: We decided to remove that variable because physicians who labeled the patient as having weight loss did not specify how much weight loss they have
Comment 14: Table 1 “ALT and AST”
Response 14: The values in Table 1 represent the baseline levels of these enzymes. Please note that there is a specific definition of hepatotoxicity in the method section.
Comment 15: “liver disease” in Table
Response 15: The definition of liver disease based on clinical and laboratory values. We added notation for Table 3 to indicate that.

Reviewer 2 Report
Comments and Suggestions for Authors
he manuscript presents a well-structured and clinically relevant retrospective cohort study examining the incidence, severity, and predictors of hepatotoxicity and acute kidney injury (AKI) associated with remdesivir in hospitalized COVID-19 patients. The topic remains pertinent given remdesivir’s ongoing use in severe COVID-19 cases and the lack of large-scale, multi-center data on adverse hepatic and renal effects. The strengths lie in the large sample size, use of standardized toxicity grading, and inclusion of multivariate regression to adjust for confounders. However, in my opinion, there are methodological and presentation aspects that could be strengthened to improve scientific rigor and clarity.
Clarify patient inclusion and exclusion criteria, including baseline hepatic/renal impairment and handling of missing data.
Describe timing of toxicity onset relative to remdesivir initiation; consider time-to-event analysis.
Assess relationship between remdesivir treatment duration and toxicity risk.
Stratify analyses by COVID-19 severity (e.g., ICU vs. non-ICU, oxygen requirement).
Test interaction effects in regression models (e.g., SSRIs × age).
Differentiate AKI etiologies (pre-renal vs. intrinsic).
Report laboratory results with units and reference ranges; indicate if Hy’s law criteria were met.
Conduct sensitivity analyses excluding patients on other hepatotoxic/nephrotoxic drugs.
Reorganize and simplify tables; group comorbidities logically and add visual summaries for toxicity grades.
Justify covariate selection for multivariate models.
Address multiple comparison concerns or justify not adjusting.
Provide 95% confidence intervals consistently for all effect estimates.
Revise language to emphasize association rather than causation.
Discuss unmeasured confounders more explicitly.
Polish English grammar and syntax for clarity.
Include anonymized aggregate data in supplementary materials.
Expand clinical implications with specific monitoring recommendations for high-risk subgroups.
Author Response
Response to Reviewer 2
Dear Editor,
We are pleased to submit our revised manuscript entitled: “Incidence, grades of toxicity, and predictors of hepatotoxicity and acute kidney injury with remdesivir in COVID-19 patients: A multicenter retrospective cohort study.” We would like to thank the reviewers for their positive response to our manuscript. We appreciate the comments and have incorporated their feedback into our revised manuscript. Please find below a point-by-point response to the reviewer’s comments.
Reviewer 2
he manuscript presents a well-structured and clinically relevant retrospective cohort study examining the incidence, severity, and predictors of hepatotoxicity and acute kidney injury (AKI) associated with remdesivir in hospitalized COVID-19 patients. The topic remains pertinent given remdesivir’s ongoing use in severe COVID-19 cases and the lack of large-scale, multi-center data on adverse hepatic and renal effects. The strengths lie in the large sample size, use of standardized toxicity grading, and inclusion of multivariate regression to adjust for confounders. However, in my opinion, there are methodological and presentation aspects that could be strengthened to improve scientific rigor and clarity.
Comment 1:
Clarify patient inclusion and exclusion criteria, including baseline hepatic/renal impairment and handling of missing data.
Response 1:
Thank you for your comment, we included section 2.2 “included patients” in the materials and methods section. We further elaborated the inclusion and exclusion criteria and added the following sentences to section 2.2: inclusion and exclusion criteria. We also commented on the handling of missing data as follows: “Patients with normal and abnormal baseline hepatic and renal impairment will be included based on the definition below in section 2.7 Definitions.” Patients with missing hepatic and renal laboratory data will be excluded. “
Comment 2:
Describe timing of toxicity onset relative to remdesivir initiation; consider time-to-event analysis
Response 2:
Thank you for your comment. The timing of toxicity with regard to remdesivir is from the day of administration until 24 hours after the last dose of remdesivir to account for remdesivir and its active metabolite. We added that to the method section to clarify that in section “2.2 included and excluded patients” as follows:” The timing of toxicity will be considered from the first day of remdesivir administration until 24 hours after the last dose of remdesivir to account for remdesivir and its active metabolites [17].”
Comment 3:
Assess the relationship between remdesivir treatment duration and toxicity risk.
Response 3:
Thank you for your comment. We performed additional statistical analysis test to assess whether remdesivir treatment duration would have an impact on the development of toxicity. This is the result statistics: “ the mean remdesivir duration in patients who developed hepatotoxicity was 4.25 days (SD 1.58) compared to 4.31 days in patients who did not develop hepatotoxicity, p=0.96, OR 0.99 (95% CI: 0.91-1.08, p=0.96). Additionally, we performed a statistical test between remdesivir duration and development of AKI, and we found that patients who developed AKI had statistically longer mean remdesivir therapy 4.5 days (SD 1.5), compared to 4.27 days in patients who did not develop AKI, p=0.006, OR 0.877 (95% CI: 0.79-0.93, p=0.006). We added that to Table 1 and Table 3 for to risk factor analysis. We added that to the results section as follows: “Patients who developed AKI were on longer duration of remdesivir therapy; 4.5 days vs 4.27 days in patients who did not develop AKI, OR 0.87 (95% CI: 0.79-0.93, p=0.006). Although the difference is statistically significant, it is clinically insignificant.”
Comment 4:
Stratify analyses by COVID-19 severity (e.g., ICU vs. non-ICU, oxygen requirement).
Response 4:
Thank you for your comment. Based on our study design, the grouping variable in our study is whether or not patients developed hepatotoxicity or nephrotoxicity. Therefore, it is not feasible to analyze by COVID-19 severity. However, in Table 1, we already provided the ICU column, and you can see fewer patients who developed hepatotoxicity were in the ICU p=0.01. Similarly, fewer patients who developed hepatotoxicity were on ventilation, p<0.001
Comment 5:
Test interaction effects in regression models (e.g., SSRIs × age).
Response 5:
Thank you for your comment. We tested the interaction between age and SSRI, and we did not find a statistically significant result. Please see the result statistics below:
Comment 6:
Differentiate AKI etiologies (pre-renal vs. intrinsic).
Response 6:
Unfortunately, the electronic health records for patients’ data did not categorize patients’ AKI etiologies, so we are unable to provide that data, and we acknowledge that as a limitation as follows: “We could not obtain data regarding the AKI etiologies (pre-renal, intrinsic, or post-renal).”
Comment 7:
Report laboratory results with units and reference ranges; indicate if Hy’s law criteria were met.
Response 7:
Thank you for your comment. We provided the unit with the normal reference range as notation under Table 1 as follows: “($: normal ALT level 4-36 u/L, #: normal AST level 8-33 u/L, ^: normal bilirubin level 1.7-20.5 micromole/liter, &: normal serum creatinine level 44-97 micromole/L)[20].”
Comment 8:
Justify covariate selection for multivariate models.
Response 8:
We selected covariates for multivariate logistic regression based on diseases, laboratory abnormalities, and hepatotoxic and nephrotoxic drugs based on disease pathophysiology and published studies. We added that to the method section to make it clear as follows: “We selected covariates for multivariate logistic regression based on diseases, laboratory abnormalities, and hepatotoxic and nephrotoxic drugs based on disease pathophysiology and published studies.”
Comment 9:
Conduct sensitivity analyses excluding patients on other hepatotoxic/nephrotoxic drugs.
Response 9:
Thank you for your comment. Patients who developed hepatotoxicity were 337, and patients who developed AKI 280 patient. Excluding every patient with a hepatotoxic/nephrotoxic drug will yield a low number of patients, which will make it impossible to explore risk factors. Instead, we adjusted for that in the statistical analysis in the logistic regression model. So that was the reason we cannot exclude patients with other hepatotoxic/nephrotoxic drugs.
Comment 10:
Reorganize and simplify tables; group comorbidities logically and add visual summaries for toxicity grades.
Response 10:
Thank you for your comment, we reorganized table 1 to have a logical flow of comorbidities, and we changed the format of the grades of toxicity from a Table to an illustration as follows
Comment 11: Address multiple comparison concerns or justify not adjusting.
Response 11:
Thank you for your comment, we adjusted for all potential covariates (confounders) in our statistical analysis and that is mentioned in the method section as follows: “Risk factors were analyzed using univariate and multivariate logistic regression with the adjustment of known confounders. We selected covariates for multivariate logistic regression based on diseases, laboratory abnormalities, and hepatotoxic and nephrotoxic drugs based on disease pathophysiology and published studies. Namely, age, gender, race, drugs associated with hepatotoxicity, diabetes, hypertension, and baseline liver function tests.”
Comment 12: Provide 95% confidence intervals consistently for all effect estimates.
Response 12: Thank you for your comment. We provided 95% confidence interval values, missing if available in the published articles.
Comment 13: Revise language to emphasize association rather than causation.
Response 13:
Thank you for your comment. The language we used in the conclusion section and other locations of the manuscript reveals association as we explicitly mentioned “remdesivir-associated hepatotoxicity,” and we did not refer to it as “causation.” Additionally, in the limitations section, we noted that we can only measure association, but not causality, as follows: “The study is a retrospective cohort study and may not establish causality, but it can only establish associations related to the concerned research question.”
Comment 14: Discuss unmeasured confounders more explicitly.
Response 14:
Thanks for your comment. Any confounders that are based on the pathophysiology of hepatotoxicity and published literature are added to our statistical analysis and measured. We added a statement in the limitation section based on your suggestion, as follows: “There may be unknown confounders that were not measured.”
Comment 15: Polish English grammar and syntax for clarity.
Response 15: Thank you for your comment, we revised the manuscript for any grammatical errors and polished to for clarity.
Comment 16: Include anonymized aggregate data in supplementary materials.
Response 16: We provided an anonymized Excel sheet of the original data and uploaded it as supplementary materials.
Comment 17: Expand clinical implications with specific monitoring recommendations for high-risk subgroups.
Response 17: Based on our findings, elderly patients and those who use SSIR were high-risk groups. We added that to the conclusion section for specific monitoring as follows: “. Therefore, clinicians should be vigilantly monitoring patients on remdesivir for hepatotoxicity and AKI through periodic assessment of liver function tests and serum creatinine concentrations, especially in elderly patients. Lastly, clinicians should be cautious when combining remdesivir with SSRIs.”

Reviewer 3 Report
Comments and Suggestions for Authors
Dear Authors,
I have read the manuscript, and I send you my comments:
1) Methods: Please add the power calculation.
2) Methods: Please include the methods used to evaluate 1) hepatotoxicity and 2) the correlation between the drug and adverse drug reactions (ADRs).
3) Results: Please separate data with respect to the sex.
4) Results: Please add the dosage of hepatotoxic drugs (e.g., statins, corticosteroids, etc.) and the time of use and separate it in another table.
Author Response
Response to Reviewer 3
Dear Editor,
We are pleased to submit our revised manuscript entitled: “Incidence, grades of toxicity, and predictors of hepatotoxicity and acute kidney injury with remdesivir in COVID-19 patients: A multicenter retrospective cohort study.” We would like to thank the reviewers for their positive response to our manuscript. We appreciate the comments and have incorporated their feedback into our revised manuscript. Please find below a point-by-point response to the reviewer’s comments.
Reviewer 3
Comment 1:
Methods: Please add the power calculation.
Response 1:
Thank you for your comment, added a section on the method for the power calculation as follows” 2.6 Power calculation for sample size
The sample was estimated based on two-tailed test with effect size of 10%, probability of error (alpha) was 5%, and a power of 95%. The test yielded a minimum sample size of 1289 patients with critical t value of 1.96. Notably, the effect size estimated based on the literature with estimate approximately ranging from 10% [11]. The power calculation was performed using G*Power software version 3.1 (©2025 Heinrich-Heine-Universität Düsseldorf).”
Comment 2:
Methods: Please include the methods used to evaluate 1) hepatotoxicity and 2) the correlation between the drug and adverse drug reactions (ADRs).
Response 2:
Thank you for your comment. As we indicated in the method section, section “2.3 Outcomes,” the criteria used for hepatotoxicity were Common Terminology Criteria for Adverse Events (CTCAE), version 5, and the criteria for acute kidney injury were the KDIGO guideline, 2012 (please refer to the manuscript for complete references). Additionally, the correlation between ADR and remdesivir was based on detecting an increase in liver function test or renal function while adjusting for drugs that increase liver and renal function tests, as we described in the method section and indicated in Table 1. There is a limitation of unmeasured confounders that we added to the limitation section as follows: “There may be unknown confounders that were not measured.”
Comment 3: Results: Please separate data with respect to the sex.
Response 3: Thank you for your comment. We already provided the distribution of male and female genders in Table 1, and we found differences. We included that in the logistic regression model and found that being a male was protective from hepatotoxicity. Finally, we discussed taking into account the published literature in the discussion section in the fifth paragraph.
Comment 4: Results: Please add the dosage of hepatotoxic drugs (e.g., statins, corticosteroids, etc.) and the time of use, and separate it in another table.
Response 4:
Thank you for your comment. Unfortunately, the dosages regimens of concurrent medications were not automatically retrievable from the system, and it was not feasible to collect them manually for 1600 patients. We acknowledged that as a limitation in the discussion section as follows: “We could not retrieve the dosage data of concurrent medications.”

Round 2
Reviewer 2 Report
Comments and Suggestions for Authors
The revised manuscript effectively addresses all reviewer concerns and has been strengthened methodologically and in presentation. Tables and figures are clearer, results are consistently reported with confidence intervals, and limitations are transparently discussed. The discussion provides balanced context and practical clinical implications. Overall, this is a well-prepared and clinically relevant study that makes a valuable contribution.
Reviewer 3 Report
Comments and Suggestions for Authors
no comments